# Cell-Free DNA Hypermethylation in Patients with Acute Pancreatitis

**DOI:** 10.3390/ijms262110792

**Published:** 2025-11-06

**Authors:** Hassan Al-Mashat, Daniel Roger Baddoo, Søren Lundbye-Christensen, Poul Henning Madsen, Inge Søkilde Pedersen, Henrik B. Krarup, Benjamin Emil Stubbe, Ole Thorlacius-Ussing, Stine Dam Henriksen

**Affiliations:** 1Department of Gastrointestinal Surgery, Aalborg University Hospital, 9000 Aalborg, Denmark; 2Research Data and Biostatistics, Aalborg University Hospital, 9000 Aalborg, Denmark; 3Department of Molecular Diagnostics, Aalborg University Hospital, 9000 Aalborg, Denmark; 4Department of Clinical Medicine, Aalborg University, 9000 Aalborg, Denmark; 5Clinical Cancer Research Center, Aalborg University Hospital, 9000 Aalborg, Denmark

**Keywords:** hypermethylation, cell-free DNA, acute pancreatitis, epigenetic, methylation

## Abstract

Cell-free DNA (cfDNA) promoter hypermethylation shows promise as a blood-based biomarker for pancreatic cancer, and similar alterations may occur in acute pancreatitis (AP). This study investigated the cfDNA hypermethylation profile of AP patients over time, compared with healthy controls, and its association with AP severity markers. A prospective longitudinal study including hospitalized AP patients and healthy controls was conducted. Methylation-specific PCR of a 23-gene panel was performed on plasma collected at inclusion (T0), 6 weeks (T6W), 6 months (T6M), and 7–8 years (T8Y). Associations between gene hypermethylation and clinical markers of AP severity—CRP, leukocyte count, creatinine, hospital stay, and complications—were evaluated. AP patients had a significantly higher mean number of hypermethylated genes at T0 (7.4, 95% CI: 6.8–8.0) compared with the controls (3.3, 95% CI: 2.8–3.8; *p* < 0.01). The mean number decreased over time to 3.2 (95% CI: 2.4–4.1) at T8Y. Total hypermethylation was positively associated with CRP (ρ = 0.39; *p* = 0.0018), leukocytes (ρ = 0.35; *p* = 0.0052), and hospital stay (ρ = 0.27; *p* = 0.0375). AP patients exhibited significantly higher cfDNA hypermethylation at disease onset, which normalized over time. Total hypermethylation showed positive associations with several markers of AP severity.

## 1. Introduction

Acute pancreatitis (AP) is a common yet unpredictable condition, with clinical trajectories ranging from mild discomfort to multi-organ failure [1,2]. Early and accurate prognostication remains a major clinical challenge [3,4,5,6]. Recent studies suggest that cell-free DNA (cfDNA) released during tissue injury may carry epigenetic signatures reflecting disease processes. Among these, DNA hypermethylation patterns have emerged as promising biomarkers in various inflammatory conditions [7,8]. This proof-of-concept study explores whether patients with AP exhibit a distinct hypermethylation profile in a selected panel of promotor genes compared to healthy controls, potentially paving the way for future studies aimed at improving diagnosis and prognostication.

Acute pancreatitis is defined as acute damage to the pancreas, often resulting in a mild self-limiting pancreatic inflammation involving the recruitment of leukocytes via several inflammatory mediators [2]. However, around 20% of patients with AP present with a moderate or severe form of the disease, resulting in a mortality rate of 20–40% [4,5]. AP is the most common pancreatic disease globally, with 34 cases per 100,000 person-years, with an increasing incidence [1].

Although improvements in diagnosis and treatment have led to decreased mortality and morbidity, AP may still result in serious complications such as diabetes, exocrine pancreatic insufficiency, and chronic pancreatitis as well as reduced quality of life [9,10]. Early detection of severe AP is essential to predict which subpopulation would benefit from closer monitoring, further medical treatment, and admission to the intensive care unit (ICU) [3,4,5,6]. Contrast-enhanced CT scans have shown high sensitivity and specificity in identifying extended pancreatic necrosis; however, these changes only become apparent days after the onset of symptoms [5]. Therefore, quick and precise, sensitive, and specific tools to predict the severity of AP are needed to identify and stratify high-risk patients. Acute pancreatitis severity is classified according to the Revised Atlanta Criteria into mild (MAP), moderately severe (MSAP), or severe AP (SAP) according to the presence of local and systemic complications and organ failure [4].

During the development of both benign and malignant pancreatic disease, several molecular changes occur, including epigenetic modifications. DNA methylation is one of several epigenetic mechanisms by which cells regulate gene expression. Changes in the methylation of the promoter regions of genes can substantially affect their expression [11,12,13]. Hypermethylation in CpG islands of the promoter region inhibits the binding of transcription factors to the DNA strand and can therefore lead to gene inactivation (e.g., silencing of tumor suppressor genes) [11,12,14]. Alterations in DNA methylation have been demonstrated in several types of cancer, including pancreatic cancer [15,16]. Changes in the methylation profile in non-malignant diseases have been studied to a lesser degree. However, several studies have found the hypermethylation of genes in chronic inflammatory diseases such as chronic obstructive pulmonary disease, helicobacter pylori-associated chronic gastritis, ulcerative colitis, virus-induced hepatitis, and rheumatoid arthritis [13,17,18].

Circulating cfDNA originates from dead cells and has been demonstrated to be in higher concentrations in both patients with cancer as well as non-malignant diseases such as sepsis, cardiovascular diseases, and traumatic brain injuries [7,8]. The methylation profile of every cell type in the body is unique, which can be used to deduce the origin and malignancy of the disease [7,8,15,19,20,21,22]. The pathophysiology of AP involves cell injury, which in turn activates a proinflammatory response by different transcription factors [2,4]. The resulting cell death contributes to the release of cfDNA, leading to higher concentrations in AP patients compared to healthy individuals [21,23,24,25]. Our group has previously investigated the methylation of 28 genes in cfDNA in patients with pancreatic ductal adenocarcinoma (PDAC) and chronic pancreatitis. However, the methylation of cfDNA has, to our knowledge, never been investigated in detail over time in patients with AP. Twenty-three of the 28 genes in the panel found in PDAC and chronic pancreatitis were also found hypermethylated in patients with AP.

Therefore, this is an exploratory study that aims to investigate the methylation profile of a selected 23-gene panel in plasma cfDNA that has previously been shown to be of relevance in pancreatic diseases. This gene panel is to be analyzed in AP patients at the time of diagnosis and compare them to healthy controls as well as to assess longitudinal changes in methylation over time in the patient group. Furthermore, this study also aims to investigate if this 23-gene panel is associated with disease severity. Ultimately, the goal is to contribute to the development of novel, cost-effective tools to improve the diagnosis and prognosis in this patient population.

## 2. Results

### 2.1. Characteristics of the Study Population

Sixty-two patients were recruited in the study period, and blood samples were collected at the time of debut of the disease (T0, *n* = 61), 4–6 weeks after (T6W, *n* = 43), approximately 6 months after (T6M, *n* = 33), and 7–8 years after (T8Y, *n* = 28). As illustrated in Figure 1, a proportion of patients were lost to follow-up. Due to severe hypotension at admission, one patient was excluded from the study. Twelve patients died in the study period. Seventy healthy controls were included in the study.

The mean age of the study population was 56 years, with roughly an equal number of males *n* = 33 (54.1%) and females *n* = 28 (45.9%). Approximately half of the patients (52.4%, *n* = 32) had gallstone pancreatitis, 14.8% (*n* = 9) had alcoholic pancreatitis, and 9.8% (*n* = 6) had a combination of the two and 23.0% (*n* = 14) had other or unknown etiology. Several patients developed complications; 11.5% (*n* = 7) of patients developed chronic pancreatitis, and 26.2% (*n* = 16) of patients had other complications post-diagnosis (several patients developed more than one complication). See Table 1 for an overview of the characteristics of the study population.

### 2.2. Change in Hypermethylation Profile of Patients over Time

A graphic representation of the change in the mean number of hypermethylated genes in patients with AP at T0, T6W, T6M, and T8Y is shown in Figure 2. This was compared to the healthy controls (*n* = 70), which acted as a baseline, since there were no data across time for this group. As is apparent from the graph, there is a clear trend in the change in hypermethylated genes over time, from approximately on average seven hypermethylated genes at the time of diagnosis and then approaching the baseline at an average of three hypermethylated genes 7–8 years after.

A repeated measures ANOVA analysis was performed (see Table 2) and revealed that patients with AP at the time of diagnosis had an average of 7.4 (95% CI: 6.8–8.0) hypermethylated genes at T0 while the healthy controls had an average of 3.3 (95% CI: 2.8–3.8). Thus, patients at the time of diagnosis with AP had more than twice the average number of hypermethylated genes compared to the healthy controls (*p* < 0.01). Likewise, patients with AP had a significantly higher mean number of hypermethylated genes at both T6W 5.3 (95% CI: 4.6–6.0) and T6M 4.9 (95% CI: 4.2–5.7) compared to the healthy controls at baseline. However, we found no statistically significant difference between patients with AP and the healthy controls at T8Y. The data were bootstrapped with 1000 repetitions, yielding similar results.

At the time of inclusion (T0), 12 of the 23 genes were significantly more frequently hypermethylated than in the healthy controls (see Appendix A). Furthermore, the percentage hypermethylation of individual genes and their trajectory across time are shown in Figure 3.

### 2.3. Clinical Correlates

#### 2.3.1. Association Between Total Hypermethylated Genes and Severity Markers in Acute Pancreatitis

The total number of hypermethylated genes in T0 and its association with the different proxy markers for AP severity is depicted in Table 3. Here, there was a positive, mild–moderate association between the number of hypermethylated genes and all the proxy markers for disease severity, except for complications. However, only CRP (ρ = 0.39, *p* = 0.002), leukocyte count (ρ = 0.35, *p* = 0.005), and length of admission (ρ = 0.27, *p* = 0.038) were statistically significant.

#### 2.3.2. Association Between Individual Hypermethylated Genes and Severity Markers in Acute Pancreatitis

All individual 23 genes were tested against the severity proxy markers using the Wilcoxon rank-sum test and Fisher’s exact test (see Table 4). Those genes that had a significant difference between the hypermethylated and non-hypermethylated groups are presented in Table 4. For information about the rest of the genes, see Appendix A. The only proxy marker that did not show any significant difference between the hypermethylated and non-hypermethylated genes was complications. The genes *HIC1*, *BRCA1*, and *RASSF1A* are significantly associated with one or more proxy markers.

#### 2.3.3. Association Between Selected Genes and Proxy Markers for Acute Pancreatitis

Based on the findings presented in Table 4, we identified seven genes that were significantly associated with proxy markers of disease severity. To assess their collective clinical relevance, this seven-gene panel was subsequently evaluated for its association with the same proxy markers (see Table 5). A moderately strong association was observed between the CRP concentration and degree of hypermethylation in the seven-gene panel (Spearman’s ρ = 0.519, *p* < 0.001). Poisson regression analysis was conducted to assess the association between the maximum CRP levels during admission and the seven-gene panel methylation count. The CRP levels were significantly associated with the methylation count (IRR = 1.002; *p* = 0.004). This signifies that for every 100-unit increase in CRP, the expected hypermethylated gene increased by approximately 17%. The model showed no evidence of overdispersion (dispersion = 0.35; *p* = 1.00).

Furthermore, there was a positive, moderately strong association between hypermethylated genes and leukocyte count in the blood (Spearman’s ρ = 0.520, *p* < 0.001). It also revealed that the leukocyte concentration was associated with the methylation count in the Poisson regression analysis (IRR = 1.031; *p* = 0.005), meaning that for every 10-unit increase in the leukocyte count, the hypermethylation count is increased by 34%. No overdispersion in the Poisson distribution was seen (dispersion = 0.35; *p* = 1.00).

A similar positive trend was found with the creatinine concentration (Spearman correlation ρ = 0.336, *p* = 0.008). Even though this showed a positive trend, no statistically significant association with the Poisson regression was found (IRR = 1.002; *p* = 0.157; dispersion = 0.46; *p* = 1.00).

As for the association between hypermethylated genes and length of admission, there was a moderately strong association (Spearman correlation ρ = 0.463, *p* < 0.001). The Poisson regression analysis showed that for every week of admission, the number of hypermethylation count increases by 5.7% (IRR = 1.008; *p* = 0.048).

When testing the seven-gene panel association with complications, logistic regression was used and no statistically significant results were found.

## 3. Discussion

### 3.1. cfDNA Hypermethylation Pattern in Patients with Acute Pancreatitis

In this present exploratory study, we examine cfDNA promoter hypermethylation in a panel of genes in AP patients and follow the hypermethylation profile over time. A significant difference was found in the total number of hypermethylated genes between AP patients at admission and healthy controls. Furthermore, we found that the number of hypermethylated genes decreased over time down to the same level as the healthy controls after eight years. Additionally, the total hypermethylated genes as well as specific individual genes were associated with proxy markers for AP severity at the time of admission. To our knowledge, this is the first study investigating hypermethylated genes over time in patients with AP. These results align with expectations based on the existing literature [13,17,18]. This trend is consistent with the dynamic nature of epigenetic modifications in response to acute inflammatory processes, where the resolution of inflammation is often accompanied by a normalization of DNA methylation patterns. These findings support the role of hypermethylation as a transient marker of disease activity in AP.

The genes examined in this current study were based on a systematic literature review by our group identifying 28 genes of interest in relation to PDAC. We detected promoter hypermethylation in cfDNA in 23 of the 28 genes in AP patients as well [15,16,26]. This led to the interest in examining the hypermethylation profile of these 23 genes over time after an AP event. Analyzing a gene panel developed for PDAC is associated with methodological concerns regarding relevance in AP. However, there are indications that molecular changes in specific genes are shared between PDAC and AP, since AP has been associated with pancreatic cancer development [27]. In addition, recurrent AP can lead to chronic pancreatitis, a risk factor associated with the development of pancreatic cancer [28]. With regard to diagnostic biomarker development for PDAC, the result of this study indicates a risk of false positive results of biomarkers based on cfDNA hypermethylation, when used in patients with AP, which is very important to be aware of in a clinical setting.

Sun and colleagues investigated whether the methylation of genes can be used to predict AP severity [29]. They performed genome-wide cfDNA methylation profiles on 61 patients with AP and 24 healthy controls in order to identify specific methylated genes of interest for AP. They identified 565 methylation haplotype blocks (MHBs) that were significantly less hypermethylated in AP patients compared to healthy controls. Furthermore, they identified 59 MHBs that were significantly less hypermethylated in SAP compared to MAP. However, the results need to be verified. In addition, further studies of different patient groups are required to conclude that the difference in methylation profiles between SAP and MAP are specific to AP and not just an illustration of a general unspecific difference in degree of inflammation. Analyzing our methylation profile according to AP severity was unfortunately not possible in this current study, as it was challenging to retrospectively determine the severity of AP according to the Revised Atlanta Criteria due to too many missing variables. However, known proxy markers for AP severity were used instead.

We also investigated the hypermethylation frequency of each individual gene in AP patients and how it changed over time compared to a baseline hypermethylation profile of healthy controls analyzed at a single timepoint. We found that there were specific genes that were hypermethylated in a higher or lower number of patients with AP than healthy controls at different timepoints. Most patients (>80%) had the genes *ESR1, MEST1v2*, and *HIC1* hypermethylated at the time of admission compared to <40% in the healthy controls (Appendix A).

### 3.2. Total Hypermethylation of 23-Gene Panel and Markers for Severity in Acute Pancreatitis

Our results describe a statistically significant association between the total number of hypermethylated genes at the time of admission (T0) and proxy markers of disease severity such as CRP, creatinine, leukocyte count, and length of admission, but not for complications. These proxy markers have previously been shown to be associated with the severity of AP [4,30,31,32]. A possible reason why we did not find a significant association between total hypermethylation at the time of admission and complications could be due to the limited number of patients with complications. Furthermore, there were only a few patients that developed severe local complications such as necrosis and more systemic complications could not be determined retrospectively due to the lack of information.

The positive association found between the total hypermethylated genes and the proxy markers for disease severity may potentially indicate that there are overlapping epigenetic changes found between the severity of AP in the acute phase and chronic pancreatitis and PDAC, which the gene panel was originally developed for.

### 3.3. Individual Hypermethylated Genes and Markers for Severity in Acute Pancreatitis

Rather than focusing solely on the overall hypermethylation status of the 23-gene panel, gene-specific analyses were made to provide more insight into the epigenetic changes that are related to disease severity. Here, we identified a gene panel of seven genes that are associated with the markers of AP severity: *APC*, *BRCA1*, *CDKN2B*, *HIC1*, *Neurog1*, *RASSF1A*, and *RARB.*

The seven-gene panel showed a positive association with proxy markers for disease severity such as CRP, leukocyte count, creatinine, and length of hospital stay. These findings from this proof-of-concept study identify seven genes of shared relevance between chronic pancreatitis/PDAC and AP. This warrants further investigation into their role in the diagnosis and prognosis of AP as well as predictors of AP severity and how these genes can be used to differentiate between AP, chronic pancreatitis, and PDAC.

### 3.4. Clinical Relevance of the Hypermethylated Genes

At the time of admission, three genes were hypermethylated in more than 80% of patients and less than 40% of healthy controls. This difference was statistically significant. These genes were *ESR1*, *MEST1v2*, and *HIC1* and they have, to our knowledge, not been previously investigated in patients with AP. They have many different functions; however, what is common between all three genes is that hypermethylation causes the decreased expression of certain proteins. When these proteins are not expressed, it promotes pro-inflammatory processes as well as tumorigenesis [33,34,35,36].

The 7 genes in the new panel formed from the 23-gene panel—*APC*, *BRCA1*, *CDKN2B*, *HIC1*, *Neurog1*, *RASSF1A*, and *RARB*—are involved in diverse biological processes. Several promote inflammation, others contribute to malignant transformation, and some participate in both [26,37,38,39,40,41,42,43,44,45]. Since AP induces an inflammatory response, it is biologically plausible that they are hypermethylated in patients with AP. However, the exact mechanisms driving the selective hypermethylation of these genes are unclear and are beyond the scope of this study.

Furthermore, it has been suggested in a Danish nationwide, population-based, matched cohort study that patients diagnosed with AP have an increased long-term risk of developing pancreatic cancer compared to matched controls [27]. Our study can possibly explain a part of the mechanism underpinning the aforementioned study’s results. This provides further grounds and justification for continuing investigating overlapping features between the three diseases, AP, chronic pancreatitis, and PDAC.

### 3.5. Limitations

The results produced from this study should be seen in light of its limitations. There are possible confounding factors contributing to epigenetic modifications that were not accounted for in the statistical analyses. Independent factors such as obesity, smoking, and aging are associated with an increase in DNA promoter hypermethylation over time [46]. The gene panel that was analyzed in this study is based on genes of interest primarily regarding PDAC, selected with the main purpose to differentiate PDAC and chronic pancreatitis and not regarding patients with AP. However, the genes were previously identified as hypermethylated in the cfDNA of AP patients as well [15,26] and the results of this study add valuable knowledge to future research in biomarker development based on DNA hypermethylation. Hence, there could be more specific gene markers for AP regarding differentiating severity, e.g., the ones identified by Sun and colleagues [29]. The lack of follow-up samples from the healthy controls at specific timepoints is also a limitation. As the methylation profile is known to change with age, it would be of great interest to know if the methylation status of the healthy controls had changed during the follow-up period of 7–8 years. In addition, the lack of clinical information about the healthy controls such as previous diseases could potentially influence the methylation profile [7,8,15,16,26,45,46]. It should also be noted that to avoid bias in a repeated measures ANOVA, the attrition of the participants should be random. In this study, nothing suggests there is a systematic attrition. To support this assumption, we investigated the data further and found that there were no significant differences in total hypermethylation at T0, CRP concentrations during admission, length of hospital admission, age and sex among patients who dropped out before eight years and those patients who completed the full length of the study. Another limitation of the study is the time gap between blood sampling at T6M and T8Y. This makes it more difficult to say exactly how long after the initial diagnosis of AP that patients keep having an increased amount of hypermethylated genes; it could be anytime between 6 months and 8 years. Moreover, one of the reasons why there was no statistically significant difference between patients and the healthy controls after 8 years could be due to a lack of statistical power based on the limited number of patients at the last follow-up (*n* = 28). Furthermore, it would be of interest to know the cause of death for the 12 patients who died during the follow-up period. Unfortunately, this information was not possible to access from the death register due to the lack of authorization from the Regional Ethics Committee.

## 4. Materials and Methods

### 4.1. Study Design and Setting

This was a prospective, longitudinal study of patients diagnosed with AP in the Department of Gastrointestinal Surgery, Aalborg University Hospital Denmark and the Department of General Surgery, Hospital of the North Denmark Region, between November 2013 and May 2015. All participants were given both oral and written information before giving their written consent prior to partaking in the study. The study was authorized by the Research Ethics Committee in the North Denmark Region (N-20130037).

### 4.2. Population and Blood Samples

The study population was patients diagnosed with AP. The diagnosis of AP was based on the presence of at least two of the following three criteria: localized pain in the epigastrium, a three-fold increase in the upper limit reference value of amylase (upper limit 65 U/L) or lipase (upper limit 60 U/L), and CT- or MRI-verified AP changes. Blood samples were obtained at the time of diagnosis (within 48 h after hospitalization), 4–6 weeks after, approximately 6 months after, and 7–8 years after initial admission. For the 7–8 years follow-up, blood samples obtained between July 2022 and October 2022, the participants had the option to deliver a blood test at or close to their residence to increase the follow-up rate. Blood donors were included as a healthy control group and consisted of unselected healthy volunteers from the North Denmark Region. Other than age and sex, no further data were available.

### 4.3. Laboratory Analysis

The methylation analysis was conducted by a senior laboratory scientist at the Department of Molecular Diagnostics, Aalborg University Hospital. All the blood tests were centrifuged at 4000 rpm at 4 °C for 20 min after which plasma was aliquoted and stored at −80 °C within two hours of sample collection. The methylated cfDNA was detected using a methylation-specific PCR technique based on bisulfite treatment. Plasma cfDNA was purified and extracted using the EasyMAG nucleic acid purification platform (Biomeriéux, Marcy-l’Étoile, France). Afterwards, extracted DNA was mixed with a deamination solution at 90 °C for 10 min. Lastly, a final DNA extraction was performed and eluted in a KOH solution [15,47].

Using Beacon Designer^®^ (PREMIER Biosoft International, Palo Alto, CA, USA), primers and probes were designed (see additional file 1 in ref. [15]) to contain many CpGs and to bind before the promoter regions of the genes (i.e., exon one). Because we expect cfDNA fragments to be about 160 base pairs, the PCR products were designed to be below 140–150 base pairs [15,48].

Two PCR amplifications were conducted. The aim of the first PCR round was to increase the amount of cfDNA present in the sample. This was performed by using outer methylation-specific primers added to the sample to test for the 23 promoter regions. After this, a second round of PCR was performed in order to analyze specific genes of interest. Inner methylation-specific primers and methylation-specific probes were used in the second round. Hemi-methylated MEST transcript variant 1 (MEST1v1 M) was used as the reference gene in both PCR amplifications. The methylation status of all 23 genes was dichotomized. The gene was considered as hypermethylated if there was a measurable ct value (independent of the value). This dichotomization has been used in previous studies [16]. For a more detailed description of the specific laboratory analyses that have been carried out, refer to Henriksen et al. 2016 [15].

### 4.4. Assessment of Acute Pancreatitis Severity

In order to investigate the clinical relevance of the gene panel, especially if it is associated with disease severity, we had to determine the severity of AP in our patient population. This was not possible to do retrospectively due to the lack of information. Therefore, we used proxy markers that are known to be associated with AP severity such as maximum CRP concentration, leukocyte count, and creatinine during the initial admission [30,31]. Additionally, the length of hospital admission and whether the patients developed complications were also used [4,32]. Complications in this study were defined as either the development of pseudocysts, chronic pancreatitis, stent placement surgery, endocrine/exocrine insufficiency, or necrosis. This was in order to determine if there is an association between the hypermethylation of the gene panel in question and disease severity, which can provide evidence for the pursuit of further investigations.

### 4.5. Statistical Analysis and the Deduction of a New Seven-Gene Panel

No power calculations were performed prior to recruiting patients for this study since the aim of this study was exploratory in nature and was not aimed at testing a specific hypothesis.

Basic descriptive information from the participants was collected. Continuous variables were described with mean and standard deviation (SD) and categorical variables as a percentage. The frequency of methylated genes for all genes in the panel for both AP patients and healthy controls was presented and the difference between the groups was tested using Fisher’s exact test.

A comparison was made between the methylation profile in AP patients at four different timepoints and the hypermethylation profile in the healthy controls at a single timepoint. In this analysis, the healthy controls act as a reference population. Furthermore, the mean number of hypermethylated genes was calculated for both groups. Changes over time in the mean number of hypermethylated genes among AP patients, as well as differences from baseline in the controls, were assessed using repeated measures ANOVA. This was to account for the within-patient correlation of the AP patients. Moreover, the data in Figure 2 and Table 2 were bootstrapped 1000 times to accommodate potential deviations from normality and variance homogeneity.

Association between the total hypermethylation status of the 23-gene panel and proxy markers for AP severity (e.g., CRP, leucocyte count, creatinine, and length of hospital admission) was tested using Spearman’s rank correlation coefficient. Test of association between the total number of hypermethylated genes and complications, which was a binary variable, was calculated using logistic regression with odds ratios and 95% confidence intervals.

Additionally, associations between the hypermethylation status of the individual 23 genes and CRP, leucocyte count, creatinine, and length of hospital admission were investigated using the Wilcoxon rank-sum test with medians and IQR calculated. As for the association between the individual 23 genes and complications, Fisher’s exact test was performed.

The significant seven genes associated with the clinical proxy markers for AP severity from the above analyses were collected and used as a new seven-gene panel. To investigate the new gene panel’s association with the proxy markers, Spearman’s rank correlation coefficient and Poisson regression analyses were calculated and performed. Logistic regression with odds ratios and 95% confidence intervals was performed when testing the association between the seven-gene panel and complications.

The use of non-parametric statistical tests was due to the fact that the outcome variables were not normally distributed, which was tested using the Shapiro–Wilk test and visual inspection of histograms.

The data were collected and managed using Research Electronic Capture (REDCap) tool (hosted at Aalborg University Hospital, Denmark, (https://redcap.rn.dk). StataCorp Stata Statistical Sofware (StataCorp LLC, Release 17, College Station, TX, USA) and RStudios (Posit team, version 2025.05.0+496). RStudio: Integrated Development Environment for R.Posit Software, PBC, Boston, MA, USA) was used for statistical analyses. A *p*-value below 0.05 (two-sided) was considered statistically significant.

## 5. Conclusions

This was an exploratory, proof-of-concept study to determine whether genes previously associated with chronic pancreatitis and pancreatic ductal adenocarcinoma exhibit similar cfDNA hypermethylation changes in patients with acute pancreatitis over time.

In conclusion, this study shows that the hypermethylation of several genes can be detected in the plasma cfDNA of AP patients. In addition, the hypermethylation profile of patients with AP is significantly different from the healthy controls at the time of diagnosis until at least 6 months after an AP event. This study illustrates a gradual reduction over time in the hypermethylation of several genes in AP patients and is normalized 7–8 years after the diagnosis. Furthermore, the 23-gene panel showed positive associations with the proxy markers for AP severity and a subset of 7 genes from the original 23 genes may hold potential clinical relevance.

The results from this study need to be verified in an external cohort. However, the current study can form a basis for further, larger, and well-designed studies.

## Figures and Tables

**Figure 1 ijms-26-10792-f001:**
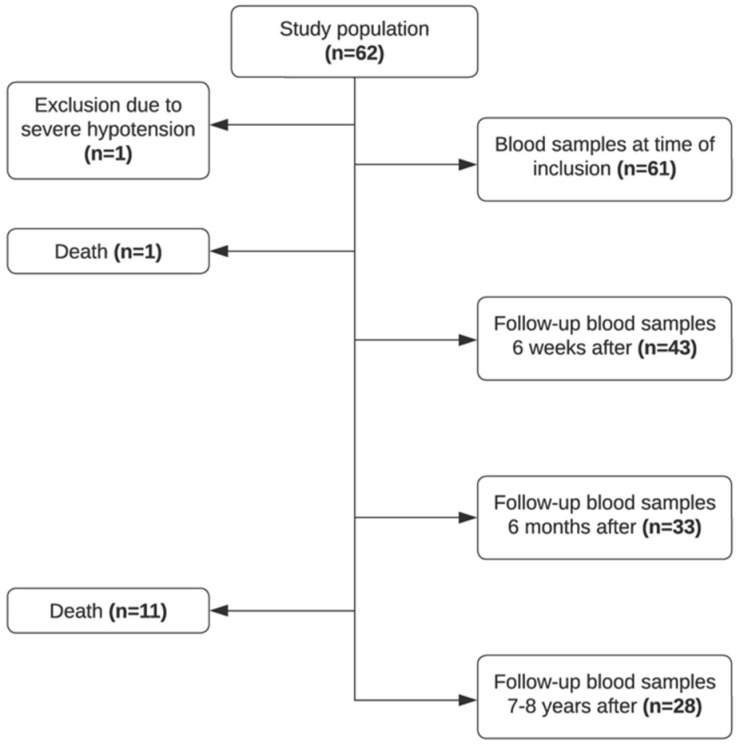
Flowchart of included patients in the study.

**Figure 2 ijms-26-10792-f002:**
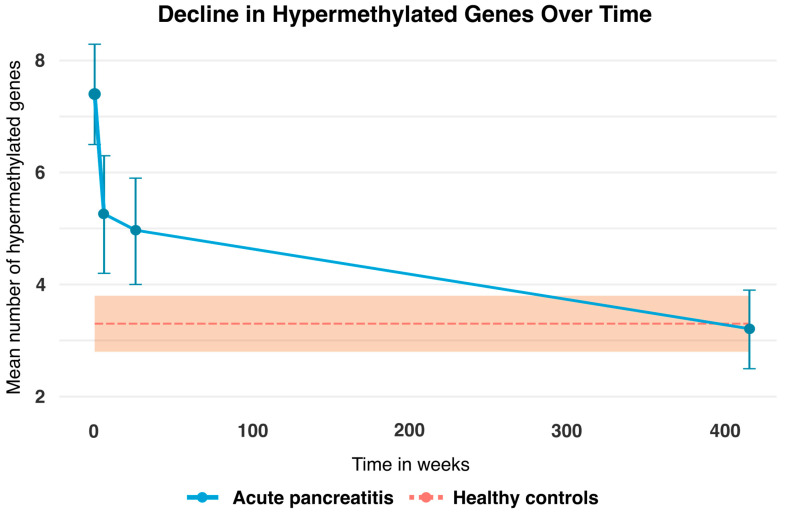
Mean number of hypermethylated genes (blue line) at T0 = time of admission was 7.4 (95% CI: 6.8–8.0), T6W = 6 weeks after admission was 5.3 (95% CI: 4.6–6.0), T6M = 6 months after admission was 4.9 (95% CI: 4.2–5.7), and T8Y = 7–8 years after admission was 3.2 (95% CI: 2.4–4.1) for patients with AP. The bars represent 95% confidence intervals for the patients. Note that the healthy controls (dotted orange line) act as a baseline and their mean hypermethylated genes were 3.3 (95% CI: 2.8–3.8). No temporal data are available for the healthy controls; only a single measurement at T8Y is included. The orange area represents the 95% confidence intervals for the healthy controls.

**Figure 3 ijms-26-10792-f003:**
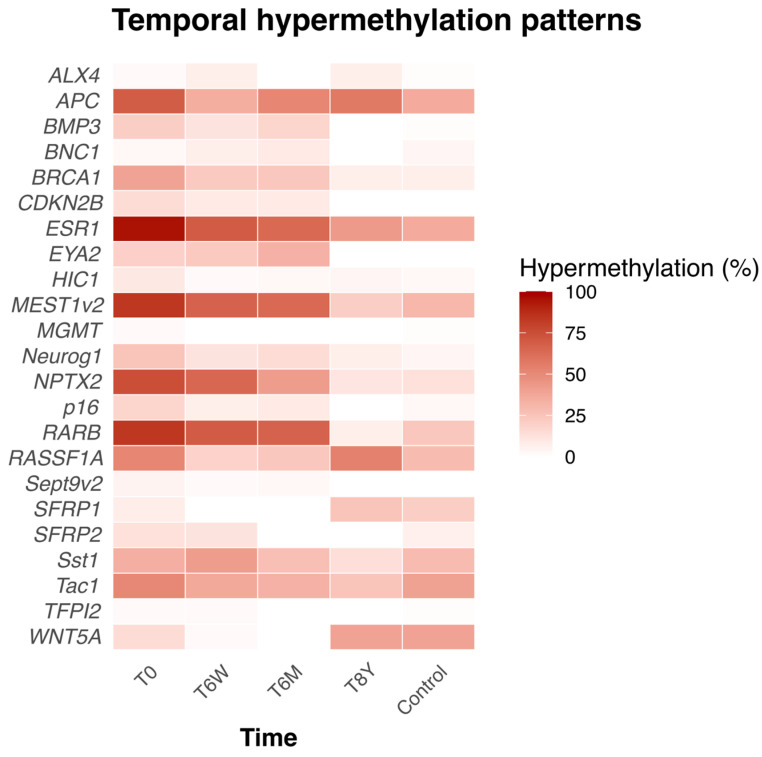
Heatmap showing the percentage of hypermethylated genes in AP patients and their trajectories over time, along with the percentage of hypermethylated genes in healthy controls at a single timepoint. T0 = at time of admission. T6W = 6 weeks after admission, T6M = 6 months after admission. T8Y = 8 years after admission.

**Table 1 ijms-26-10792-t001:** Characteristics of the study population and control population.

	Acute Pancreatitis		Controls		
	*n*	Mean (SD) or %	*n*	Mean (SD) or %	*p*-Value
**Age**	61	56 (17)	70	48.2 (12.9)	<0.01 *
**Sex**					
Male	33	54.1%	30	42.9%	0.20 **
Female	28	45.9%	40	57.1%	
**Mortality**	12	19.7%			
**BMI**	45	26.4 (4.9)			
**Smoking**					
Non-smoker	23	37.7%			
History of smoking	12	19.7%			
Currently smoking	24	39.3%			
Unknown	2	3.3			
**Alcohol consumption**					
Within recommendations	43	70.5%			
Previous overconsumption	2	3.3%			
Overconsumption	15	24.6%			
Unknown	1	1.6			
**Type of AP**					
Interstitial edematous pancreatitis	21	33.9%			
Necrotizing pancreatitis	2	3.2%			
**Comorbidities**	22	36.1%			
T1DM	1	1.6%			
Alzheimer’s disease	2	3.3%			
Depression	1	1.6%			
Stroke	7	11.5%			
Cancer pre-inclusion	7	11.5%			
Cancer post-inclusion	8	13.1%			
Chronic pancreatitis	6	9.8%			
**Complications**	23	47.5%			
Pseudocysts	7	11.5%			
Chronic pancreatitis	7	11.5%			
Stent placement surgery	7	11.5%			
Necrosis	2	3.3%			

The definitions of alcohol consumption are based on the previous definitions presented by the National Danish Board of Health as follows: ‘within recommendations’ (<7 and <14 units/week for females and males, respectively) and overconsumption (>7 and >14 units/week for females and males, respectively). The included comorbidities are known to increase the cfDNA concentration in the blood (type 1 diabetes mellitus, Alzheimer’s disease, depression, stroke, cancers of any type before and after inclusion, and chronic pancreatitis). * Two-sample *t*-test. ** Pearson Chi-squared test.

**Table 2 ijms-26-10792-t002:** Repeated measures ANOVA fitted for patients with AP at T0, T6W, T6M, and T8Y and comparing these to the healthy controls, which act as a baseline.

Time	Mean Hypermethylated Genes (95% CI)		*p*-Values
	AP	Healthy Controls	
0	7.4 (6.8–8.0)	3.3 (2.8–3.8)	*p* < 0.01
6 weeks	5.3 (4.6–6.0)		*p* < 0.01
6 months	4.9 (4.2–5.7)		*p* < 0.01
8 years	3.2 (2.4–4.1)		*p* = 0.90

Note that there is only a single measurement for the healthy controls at one timepoint. That single measurement is compared to the AP group at all timepoints. Mean and 95% confidence intervals as well as *p*-values are presented.

**Table 3 ijms-26-10792-t003:** Depicts the association between total number of hypermethylated genes in the 23-gene panel at time of admission (T0) and different proxy markers for AP severity.

Proxy Marker	Association (ρ or OR)	*p*-Value
CRP ^a^	ρ = 0.39	0.002
Leukocytes ^a^	ρ = 0.35	0.005
Creatinine ^a^	ρ = 0.21	0.101
Length of admission (days) ^a^	ρ = 0.27	0.038
Complications ^b^	OR = 1.07	0.520

^a^ Spearman’s rank correlation. ^b^ Logistic regression model with odds ratio (OR).

**Table 4 ijms-26-10792-t004:** Median and IQR values of selected genes at admission (T0) in hypermethylated versus non-hypermethylated groups, stratified by proxy markers for AP severity.

Proxy Marker	Genes	Hypermethylated Group, Median (IQR)	Non-Hypermethylated Group, Median (IQR)	*p*-Value
CRP (mg/L) ^a^	*APC*	173 (98–254)	46 (15–193)	0.008
	*HIC1*	293 (224–390)	146 (48–216)	0.034
	*Neurog1*	193 (76–265)	81 (26–145)	0.016
	*RARB*	169 (70–262)	63 (21–130)	0.019
Leukocytes (×10^9^/L) ^a^	*CDKN2B*	20 (12–22)	11 (8–16)	0.028
	*HIC1*	20 (17–24)	11 (8–16)	0.005
Creatinine (µmol/L) ^a^	*BRCA1*	84 (68–116)	64 (60–86)	0.035
	*RASSF1A*	86 (63–105)	64 (60–82)	0.041
Length of admission (days) ^a^	*BRCA1*	8 (5–11)	5 (4–7)	0.006
	*HIC1*	11 (9–13)	6 (4–8)	0.007
	*RASSF1A*	7 (5–9)	5 (4–7)	0.018
Complications (yes/no) ^b^	No statistically significant genes identified			

Only genes that were statistically significant are included in the table. For visualization of the association of the hypermethylation status for all genes and the proxy markers for disease severity, see Appendix A. ^a^ Wilcoxon rank-sum test. ^b^ Fisher’s exact test.

**Table 5 ijms-26-10792-t005:** Showing the association between the seven-gene panel hypermethylation status at the time of admission (T0) and proxy markers for disease severity.

Proxy Marker	Spearman Correlation *ρ*-Value	*p*-Value	Poisson Regression IRR	*p*-Value
CRP (mg/L)	0.519	<0.001	1.002	0.004
Leukocytes (×10^9^/L)	0.520	<0.001	1.031	0.005
Creatinine (µmol/L)	0.336	0.008	1.002	0.157
Length of admission (days)	0.463	<0.001	1.008	0.048
Complications (yes/no)	No statistically significant results			

Spearman correlation coefficients ρ-values and *p*-values are presented. Poisson regression analysis with Incidence Rate Ratio (IRR) and *p*-values are also depicted. Not all decimals are shown.

## Data Availability

The original contributions presented in this study are included in the article/Appendix A. Further inquiries can be directed to the corresponding author.

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
