# Peer review of "Cell-Free DNA Hypermethylation in Patients with Acute Pancreatitis"

_ijms, 2025, doi:10.3390/ijms262110792_

Round 1
Reviewer 1 Report
Comments and Suggestions for Authors
This study presents a valuable longitudinal dataset spanning several years, which, if analyzed appropriately, could provide important insights for clinical diagnosis and disease monitoring in acute pancreatitis (AP). The effort to collect and maintain such a cohort over time is commendable.
However, the data as currently presented do not appear to provide sufficient mechanistic or clinically meaningful interpretation. While I understand the difficulty in retrospectively determining the disease stage for each patient, the authors could strengthen the analysis by illustrating the temporal dynamics of promoter hypermethylation for individual genes. Plotting the trajectories of methylation changes over time for each gene (or representative genes) would offer a clearer picture than simply reporting the percentage of hypermethylated genes or the proportion of patients carrying specific methylation events at each time point.
Although the inclusion of long-term follow-up samples (e.g., AP-T8Y) demonstrates substantial effort, their contribution to the main findings appears limited, given that methylation patterns at this time point are largely indistinguishable from those of blood donors. By that time, most patients have likely returned to a normal physiological state, except for those who did not survive and were therefore excluded. From a biological and diagnostic standpoint, it might be more informative to analyze samples collected closer to critical clinical outcomes—particularly those from patients with severe or non-resolving disease—rather than from long-term recovery stages.
Reviewer 2 Report
Comments and Suggestions for Authors
The manuscript addresses a relevant and timely topic concerning the hypermethylation of circulating free DNA (cfDNA) in patients with acute pancreatitis, proposing a novel epigenetic approach to understanding the molecular dynamics of the disease.
The study is well structured, features a rare prospective and longitudinal design, and employs appropriate methodologies based on methylation-specific PCR.
The work would benefit from a more in-depth analysis and clinical contextualization of the findings. It is recommended to strengthen the discussion on the diagnostic and prognostic applicability of these markers, particularly considering their potential differential use in pancreatic cancer.
- The methodology is solid, but it is advisable to justify the sample size, detail the management of follow-up losses, and specify whether randomness was evaluated, given the potential for attrition bias.
- It would be useful to consider the influence of variables such as age, obesity, or smoking on hypermethylation levels, as these are recognized factors in epigenetic modification.
- It is also advisable to include confidence intervals or standard deviations in the figures, describe the biological relevance of the most frequently hypermethylated genes, and, as far as possible, explore the relationship between the clinical severity of pancreatitis and the number of genes affected.
- In the discussion and conclusions, it is suggested to emphasize the exploratory nature of the study, the need for external validation, and caution regarding possible false positives when using these markers in diagnostic contexts related to pancreatic cancer.
- It is recommended to standardize abbreviations (PDAC, cfDNA, AP), improve the graphic quality of Figure 2, and restructure the abstract to more accurately highlight the objective, methodology, main results, and limitations of the study.
Author Response
Response to reviewer 2
We thank the reviewer for the constructive and thoughtful comments, which have significantly improved the manuscript. Our primary goal has been to enhance the clinical relevance and interpretability of the findings, particularly by focusing on timepoints and analyses that are more closely tied to disease activity and outcome.
As part of this revision, we have:
- Improved data visualization to better reflect temporal dynamics in promoter hypermethylation.
- Added new analyses linking hypermethylation at admission (T0) to clinical outcome measures.
- Revised several sections of the Results, Discussion, and Figures/Tables to clearly convey findings that may have diagnostic or prognostic utility.
We hope that these changes more effectively highlight the potential of cell-free DNA methylation markers in the monitoring and risk stratification of patients with acute pancreatitis. We would also like to emphasize that this study is exploratory in nature, with the primary aim of contributing to the development of hypermethylated cell-free DNA as a potential biomarker tool in acute pancreatitis. By investigating a pre-defined panel of genes previously shown to be relevant in other pancreatic diseases, we sought to identify methylation signals with potential biological and clinical significance in the context of acute inflammation. We believe that such early-stage, hypothesis-generating work is essential to guide future studies, support biomarker validation, and ultimately facilitate translation into clinical practice.
Below we address each point in detail and outline the changes we have made in the revised version.
Reviewer comment 1:
” The work would benefit from a more in-depth analysis and clinical contextualization of the findings. It is recommended to strengthen the discussion on the diagnostic and prognostic applicability of these markers, particularly considering their potential differential use in pancreatic cancer.”
We thank the reviewer for this constructive comment. In response, we have strengthened the result and discussion section by more clearly addressing the diagnostic and prognostic applicability of hypermethylated cfDNA markers in acute pancreatitis. In the present study, however, we did not directly compare the methylation profiles of acute pancreatitis with those observed in pancreatic cancer, as the primary aim was to characterize hypermethylation patterns specifically in acute pancreatitis.
Nevertheless, we fully agree that such comparisons are essential to determine the differential diagnostic potential of these markers. Although, we plan to explore this in future studies, where hypermethylation profiles in cfDNA from patients with pancreatic cancer, chronic pancreatitis, and other inflammatory conditions will be directly compared to those observed in acute pancreatitis. These efforts aim to investigate whether cfDNA methylation signatures can aid not only in monitoring disease severity and progression, but also in differentiating between inflammatory and neoplastic pancreatic diseases.
Reviewer comment 2:
“The methodology is solid, but it is advisable to justify the sample size, detail the management of follow-up losses, and specify whether randomness was evaluated, given the potential for attrition bias.”
We thank the reviewer for this valuable comment. The sample size in the present study reflects all consecutively enrolled patients who fulfilled the inclusion criteria during the study period, and as such, the study was not powered for hypothesis-testing but designed as an exploratory cohort to describe methylation patterns over time. We have now clarified this rationale in the Methods section.
To address the concern regarding potential attrition bias, we previously performed an assessment of potential differences of the hypermethylation profiles between those who dropped out after six weeks and those who stayed after that. After appreciated feed-back from the reviewer, we have performed an additional assessment of baseline characteristics between patients who completed follow-up and those who were lost to follow-up at the eight-year timepoint. No systematic differences were observed in the mean number of hypermethylated genes at T0, age, sex, maximum CRP during admission, and length of hospital stay, suggesting that the attrition was likely not selective. This now described in the revised limitations section.
Reviewer comment 3:
”It would be useful to consider the influence of variables such as age, obesity, or smoking on hypermethylation levels, as these are recognized factors in epigenetic modification”
We acknowledge the reviewer’s important point that variables such as age, obesity, and smoking are well-known factors influencing epigenetic modifications, including DNA methylation levels. However, we would like to emphasize that the present study is exploratory in nature and was designed with the primary objective of describing the temporal dynamics of cfDNA hypermethylation in patients with acute pancreatitis, rather than to model the precise determinants of methylation.
We agree that the role of age, obesity, and smoking should be considered in future analyses. We plan to conduct larger, prospectively designed studies where we can systematically collect these covariates and perform adjusted analyses to better understand their influence on cfDNA methylation in pancreatic disease and systemic inflammation.
However, there are several reasons why these covariates were not included in the current analysis. Firstly, the available dataset, though valuable, is relatively small and was not powered for multivariable analyses. Including multiple covariates such as age, BMI, and smoking status would likely increase the risk of model overfitting and reduce the reliability of the findings. Also, for key covariates such as smoking status and BMI, data were not consistently available across all patients and timepoints. Including these variables would have resulted in substantial loss of cases and statistical power. Lastly, this study aims to explore methylation patterns in a clinically acute and dynamic setting, our goal was not to establish causal relationships, but to describe patterns that may inform future hypothesis-driven research. We believe that confounder adjustment is more appropriate in follow-up studies specifically designed to address those causal links.
Reviewer comment 4:
”It is also advisable to include confidence intervals or standard deviations in the figures, describe the biological relevance of the most frequently hypermethylated genes, and, as far as possible, explore the relationship between the clinical severity of pancreatitis and the number of genes affected.”
We have re revised the relevant figures and tables to include measures of variability (e.g., confidence intervals and interquartile ranges) where appropriate.
The biological relevance of selected hypermethylated genes has been a main focus of our revisions. Although the dataset did not allow for the use of validated severity scoring systems (due to missing data, especially arterial blood gases), we have now explored associations between the total number of hypermethylated genes at admission (T0) and several proxy markers of clinical severity (CRP, creatinine, length of stay, and the occurrence of complications). These analyses are presented in the revised results section and support a possible link between the degree of cfDNA methylation and disease burden.
We have also expanded the Discussion section to include a more detailed description of the biological functions of the most frequently hypermethylated genes and those that are associated with proxy-markers for severity observed in our cohort. Specifically, we highlight several genes, which are known to be involved in processes such as inflammation, apoptosis, or pancreatic tissue remodeling, and discuss their potential relevance in the context of acute pancreatitis.
Reviewer comment 4:
”In the discussion and conclusions, it is suggested to emphasize the exploratory nature of the study, the need for external validation, and caution regarding possible false positives when using these markers in diagnostic contexts related to pancreatic cancer”
We have revised both the Discussion and Conclusion sections to better emphasize the exploratory nature of our study. As this is a hypothesis-generating analysis, we acknowledge that the findings should be interpreted with caution. We now explicitly state the need for external validation in independent cohorts.
Reviewer comment 5:
”It is recommended to standardize abbreviations (PDAC, cfDNA, AP), improve the graphic quality of Figure 2, and restructure the abstract to more accurately highlight the objective, methodology, main results, and limitations of the study.”
In response, we have now standardized all abbreviations throughout the manuscript. Additionally, the graphical quality of Figure 2 has been improved to ensure higher resolution and better readability. Finally, the abstract has been restructured to more accurately reflect the study’s objective, methodology, key findings, and limitations.

Round 2
Reviewer 1 Report
Comments and Suggestions for Authors
This revised manuscript shows clear improvement compared with the initial version and appears suitable for publication. Only a few minor points require attention:
1.Table 1: Please add the “%” symbol to both 42.9 and 57.1.
2.Lines 181–182: The figure legend currently reads, “There is no temporal data for the healthy controls only from a single measurement at T8Y.” It should be revised to: “No temporal data are available for the healthy controls; only a single measurement at T8Y is included.”
3.Lines 241–242: Please revise the legend for Figure 3 to: “Heatmap showing the percentage of hypermethylated genes in AP patients and their trajectories over time, along with the percentage of hypermethylated genes in healthy controls.”
4.The present tense is typically used to describe and discuss experimental results. The author should revise all relevant descriptions written in the past tense to the present tense.
With these minor adjustments, the manuscript will be ready for publication.
Author Response
Respone to reviewer 1
We thank the reviewer for their positive comments and feedback for the revised manuscript which we agree have improved the initial manuscript.
We have made the necessary improvements before the article can be ready for publishing and addressing point-by-point in the following:
Reviewer comment 1
“1. Table 1: Please add the “%” symbol to both 42.9 and 57.1.”
This has now been corrected. Please see table 1.
Reviewer comment 2
“2. Lines 181–182: The figure legend currently reads, “There is no temporal data for the healthy controls only from a single measurement at T8Y.” It should be revised to: “No temporal data are available for the healthy controls; only a single measurement at T8Y is included.””
This has now been corrected. Please see figure 2.
Reviewer comment 3
“3. Lines 241–242: Please revise the legend for Figure 3 to: “Heatmap showing the percentage of hypermethylated genes in AP patients and their trajectories over time, along with the percentage of hypermethylated genes in healthy controls.””
This has now been corrected. Please see figure 3.
Reviewer comment 4
“4. The present tense is typically used to describe and discuss experimental results. The author should revise all relevant descriptions written in the past tense to the present tense.”
We thank the reviewer for this valuable feedback and for suggesting improvements in the language of the manuscript. We agree that clarity and consistency of tense enhance readability. We would like to kindly ask for clarification regarding this point. Does the reviewer refer specifically to the results described in the Results and Discussion sections being revised from past- to present tense? Our previous experience is to write in the past tense when referring to the results found in the study, however we may not be familiar with the IJMS conventions- hence the question
